# High-Order Neural-Network-Based Multi-Model Nonlinear Adaptive Decoupling Control for Microclimate Environment of Plant Factory

**DOI:** 10.3390/s23198323

**Published:** 2023-10-08

**Authors:** Yonggang Wang, Ziqi Chen, Yingchun Jiang, Tan Liu

**Affiliations:** 1College of Information and Electronic Engineering, Shenyang Agricultural University, Shenyang 110866, China; wygvern@syau.edu.cn (Y.W.); liutan0822@syau.edu.cn (T.L.); 2College of Engineering, Shenyang Agricultural University, Shenyang 110866, China; jyclg@syau.edu.cn

**Keywords:** smart agriculture, plant factory, environmental control, adaptive decoupling controller, high-order neural network

## Abstract

Plant factory is an important field of practice in smart agriculture which uses highly sophisticated equipment for precision regulation of the environment to ensure crop growth and development efficiently. Environmental factors, such as temperature and humidity, significantly impact crop production in a plant factory. Given the inherent complexities of dynamic models associated with plant factory environments, including strong coupling, strong nonlinearity and multi-disturbances, a nonlinear adaptive decoupling control approach utilizing a high-order neural network is proposed which consists of a linear decoupling controller, a nonlinear decoupling controller and a switching function. In this paper, the parameters of the controller depend on the generalized minimum variance control rate, and an adaptive algorithm is presented to deal with uncertainties in the system. In addition, a high-order neural network is utilized to estimate the unmolded nonlinear terms, consequently mitigating the impact of nonlinearity on the system. The simulation results show that the mean error and standard error of the traditional controller for temperature control are 0.3615 and 0.8425, respectively. In contrast, the proposed control strategy has made significant improvements in both indicators, with results of 0.1655 and 0.6665, respectively. For humidity control, the mean error and standard error of the traditional controller are 0.1475 and 0.441, respectively. In comparison, the proposed control strategy has greatly improved on both indicators, with results of 0.0221 and 0.1541, respectively. The above results indicate that even under complex conditions, the proposed control strategy is capable of enabling the system to quickly track set values and enhance control performance. Overall, precise temperature and humidity control in plant factories and smart agriculture can enhance production efficiency, product quality and resource utilization.

## 1. Introduction

The technologies and concepts of smart agriculture provide support and impetus to plant factory technology. Plant factory is a specific application form of smart agriculture that creates a suitable microclimate environment for crop growth through enclosed cultivation systems, enabling year-round cultivation. The factors of microclimate environment include the temperature, humidity, light and carbon dioxide concentration [1,2,3,4]. Among these factors, to some extent, temperature and humidity play crucial roles for the efficient growth of crops [5,6,7]. As for these two factors, temperature directly affects the vital activity of the crops, and humidity affects the transpiration level of the crops. Therefore, it is very necessary to achieve the controllability of environmental factors, specifically maintaining appropriate temperature and humidity conditions, as this contributes to achieving high yield and quality for various crops cultivated [8,9]. However, from a control perspective, the dynamic models of environment exhibit inherent complexity characteristics by strong coupling, strong nonlinearity and multi-disturbances. As a result, conventional control methods often struggle to meet the actual requirements of a plant factory. To address the aforementioned challenges, this study investigates a dynamic microclimate model and explores an advanced control approach to achieve precise control for environmental factors.

In recent years, a few scholars have developed the internal microclimate environment model of plant factory. The development of the temperature model utilizes a simplified model derived from thermodynamic equations [8]. An accurate temperature model of plant factory was presented based on energy balance and identified using the experimental data [9]. The large-leaf equation was initially utilized for establishing a crop transpiration model. Simultaneously, it was proposed that crop transpiration serves as the primary factor influencing the humidity balance [10]. The dynamic model was presented, utilizing an HVAC system featuring a steam-powered humidifier and a hot/cold water-based heater [11]. The model of temperature, humidity and carbon dioxide concentration was studied from the perspective of plant physiology and thermodynamics based on the existing research results of plant factories [12]. It should be noted that the abovementioned dynamic model of plant factory is not comprehensive and has a limited application scope. Therefore, describing the dynamic model of humidity and temperature in the plant factory under different environmental conditions poses a significant challenge.

Over the past years, a large number of scholars have put forward advanced control methods, including adaptive control [13,14], robust control [15,16], optimal control [17,18], fuzzy control [19,20,21], feedforward control [22,23] and so on. However, it is frustrating that these controllers belong to linear controllers, so they can achieve good control effects for some linear systems or some weak nonlinear systems. As a matter of fact, the temperature and humidity system of the plant factory exhibits inherent complexity characteristics by strong nonlinearity. Therefore, there still exists a series of problems in applying traditional control strategies to the environmental control of plant factories. The reasons for the above problems are as follows:(1)The temperature and humidity system inside the plant factory is a nonlinear dynamic system. Moreover, there is a cross effect and mutual coupling between these two factors.(2)The crop itself strongly interacts with the environment. The air moisture content in the plant factory is primarily determined by the transpiration of crops. Therefore, the humidity regulation is severely influenced by crops’ transpiration resulting in deterioration of control performance.(3)The uncertainty of parameters, such as the material properties of the plant factory, can change due to significant variations in the external environment, which could affect the indoor temperature and humidity of plant factory. Moreover, during the crop cultivation process, the multi-disturbances, especially factors of external surroundings, could enter the factory via the fresh air system or personnel entering/leaving, thus disrupting the control effect of the system.

Aiming at the comprehensive complex characteristics of strong nonlinearity, parameter uncertainty and strong coupling, the introduction of neural network into controller design has received extensive attention. A new adaptive sliding-mode controller which can be used in a single-input–single-output (SISO) nonlinear control system was designed [24]. An adaptive fuzzy output constrained control design method is proposed for multi-input–multi-output (MIMO), stochastic, non-strict feedback, nonlinear systems [25]. To address the input constraints of discrete nonlinear systems, an adaptive fuzzy control method was proposed based on an observer [26]. Two neural networks are combined into an adaptive controller for the purpose of resolving the nonlinear characteristics of the system control input and the dynamic uncertainty of the model [27]. With the utilization of neural networks, a nonlinear adaptive decoupling controller was designed to improve the evaporation efficiency of the system [28]. Introducing the radial basis function (RBF) network into the controller was designed [29]. After the introduction of the RBF neural network, a new switching strategy was integrated [30].

Given the aforementioned issues and building upon existing research results, four scenarios, namely, refrigeration, heating, dehumidification, and humidification, are modeled and applied to various environmental conditions in the plant factory. After establishing the plant factory model, a nonlinear adaptive decoupling control method using high-order neural network is proposed to address the challenges associated with temperature and humidity control in plant factories. The primary contributions are outlined as follows.
(1)This paper enriches the model to enhance its realism by developing a comprehensive environmental model that captures the possible conditions that can occur in plant factories. Building upon this foundation, the different environmental conditions in the plant factory are fully considered, making it closer to reality.(2)As far as we know, there is almost no research on applying the nonlinear adaptive decoupling control method using high-order neural network to plant factory. In this paper, with the aim of improving the dynamic performance of the system while ensuring system stability, a method is proposed that combines a linear adaptive decoupling controller and a neural-network-based nonlinear adaptive decoupling controller by using a switching mechanism. The generalized minimum variance adjustment rate is used to design the generalized minimum variance controller, and the projection algorithm with dead zone is used to identify the controller parameters to achieve an adaptive system. To estimate the nonlinear term that has not been mathematically modeled in the system, the paper makes use of the strong learning ability of high-order neural networks.

## 2. System Description

### 2.1. Structural Design of Plant Factories

This paper focuses on a closed artificial light plant factory located in Liaoning province, China. The structure of this plant factory is made up of PVC panels and has a total volume of 16 m × 6 m × 3 m. Figure 1 shows the architecture of the plant factory, which includes a cultivation rack consisting of six columns of 14 m × 0.2 m × 0.2 m each. Instead of natural light, the plants in the factory are illuminated with LED lamps, the activation of which is adjusted according to the light needs of the crops. To prevent any risk of burning or scalding, the cultivation racks are positioned at a safe distance of 1.5 m away from the LED light source for crop cultivation. Additionally, a high-throughput plant imaging system measures the leaf area of the crops. 

The regulation of the plant factory environment is divided into four modes: heating/humidification, heating/dehumidification, cooling/dehumidification and cooling/humidification. Two air conditioners, each with a power of 5.384 kW, are suspended at the top of the plant factory. To control humidity, one dehumidifier and one humidifier are located inside the factory. The temperature control system primarily relies on the air conditioning system, which regulates the temperature by using a four-way valve to switch the operating mode of the condenser and evaporator. Additionally, the ventilation volume of the air conditioning system is adjusted to further control the temperature. The temperature and humidity sensors are placed at a height of 1.5 m above the factory floor to measure the necessary experimental data. The internal environmental diagram of the plant factory is shown in Figure 2.

### 2.2. Description of the Plant Factory Model

A schematic diagram of the energy balance of the plant factory has been drawn according to various dynamic environmental conditions that may be encountered in the plant factory, as shown in Figure 3. Inspired by the literature [31] and based on the first law of thermodynamics, the heat balance and humidity balance of the plant factory are established and expressed by Equations (1) and (2).
(1)$$\frac{dT_{a}}{dt}=\frac{Q_{light}+Q_{eq}+Q_{H}+Q_{hw}+Q_{v}-Q_{plant}-Q_{ca}}{\rho_{a}\cdot V\cdot C_{ap}}$$
(2)$$\frac{dH_{a}}{dt}=\frac{W_{crops}+W_{hum}+W_{ac}+W_{v}-W_{dh}}{\rho_{a}\cdot V}$$

In energy balance Equations (1) and (2), $T_{a}$(°C) is the internal temperature of the plant factory; $Q_{light}(kJ/m^{2})$ is the energy generated for artificial light sources; $Q_{eq}(kJ/m^{2})$ is the energy generated by the humidifier and dehumidifier during operation in the plant factory; $Q_{H}(kJ/m^{2})$ is the energy exchanged between the crop canopy and the air; $Q_{hw}(kJ/m^{2})$ represents the energy brought by the hot air entering the plant factory from the air conditioning system; $Q_{v}(kJ/m^{2})$ is the energy brought in (or taken out) by the new air ventilation system; $Q_{plant}(kJ/m^{2})$ is the energy consumed by crop transpiration; $Q_{ca}(kJ/m^{2})$ is the energy generated to the plant factory when the air conditioner delivers cold air; $\rho_{a}(kg/m^{3})$ is the density of air in a plant factory; $V(m^{3})$ is the internal volume of a plant factory; ${\;C}_{ap}(kJ/(kg\cdot {}^{\circ}C))$ is the specific heat at constant pressure in a plant factory; $H_{a}(kg/kg)$ is the humidity ratio of the plant factory; $W_{plant}(kg/s)$ is the impact of crop transpiration on internal humidity, mainly represented by evaporation of crops; $W_{ac}(kg/s)$ is the moisture content of the air supplied by the air conditioning; $W_{hum}(kg/s)$ is the moisture content of the air supplied by the humidifier; ${\;W}_{v}(kg/s)$ is the moisture content of the outside air entering the plant factory during ventilation; $W_{dh}(kg/s)$ is the moisture content of air supplied by the dehumidifier.
(1)Heat balance equation for artificial light sources

This paper uses LED lamps as artificial light sources. The energy of artificial light sources mainly enters the plant factory in the form of light radiation and heat dissipation. The plant canopy absorbs light radiation, which is then converted into thermal energy. Additionally, the heat dissipation of the light source can contribute to a slight increase in the temperature within the plant factory. According to references [9,32], establishing the energy balance equation for artificial light sources is expressed by Equation (3).
(3)$${Q_{light}=Q}_{diss}+Q_{rad}=k_{1}\cdot k_{2}\sum_{i=1}^{n}P_{i}\cdot t_{w}$$
where $Q_{diss}(kJ/m^{2})$ is the energy emitted by the LED lamps; $Q_{rad}(kJ/m^{2})$ is the energy radiated by the LED lamps; $k_{1}$ is the lighting utilization factor of LED lamps; $k_{2}$ is the special compensation factor of LED lamps; n is the total number of LED lamps in the plant factory; $P_{i}(w)$ is the power of a single LED lamp; $t_{w}(h)$ is the working time of LED lamps.
(2)Heat balance equation for dehumidifier and humidifier

$Q_{eq}$ is composed of $Q_{dh}$ and $Q_{hum}$, as shown in Equation (4). The heat generated by the dehumidifier ($Q_{dh}$) and humidifier ($Q_{hum}$) during their operation is represented by Equations (5) and (6).
(4)$$Q_{eq}=Q_{dh}+Q_{hum}$$
(5)$$Q_{dh}=\rho_{a}\cdot C_{ap}\cdot q_{dh}\cdot (T_{dh}-T_{a})$$
(6)$$Q_{hum}=\rho_{a}\cdot C_{ap}\cdot q_{hum}\cdot (T_{hum}-T_{a})$$
where $Q_{dh}(kJ/m^{2})$ is the energy generated by the dehumidifier during operation; $Q_{hum}(kJ/m^{2})$ is the energy generated by the humidifier during operation; $q_{dh}(m^{3}/s)$ is the air flow rate of the dehumidifier; $T_{dh}$(°C) represents the heat generated during the operation of the dehumidifier; $q_{hum}(m^{3}/s)$ is the air flow rate of the humidifier; $T_{hum}$(°C) represents the heat generated during the operation of the humidifier.
(3)Crop canopy heat balance equation

According to Fick’s first diffusion law [33], the establishment of the crop canopy heat balance equation is represented by Equation (7):(7)$$Q_{H}=A\cdot H=A\cdot LAI\cdot \rho_{a}\cdot C_{ap}\frac{\left(T_{p}-T_{a}\right)}{r_{a}}=A_{P}\cdot \rho_{a}\cdot C_{ap}\frac{\left(T_{p}-T_{a}\right)}{r_{a}}$$
where $A(m^{2})$ is the internal ground area of a plant factory; $H(kJ/m^{2})$ is the crop canopy and air sensible heat exchange; LAI is the leaf area index of the crop; $T_{p}$(°C) is the temperature of the crop; $A_{p}(m^{2})$ is the leaf area of the crop; $r_{a}$ is the boundary layer dynamic resistance of crop leaves.
(4)Heat balance equation for conveying hot air

This article mainly uses air conditioning to achieve indoor temperature increase. The heat balance equation for conveying hot air is established, which is represented by Equation (8).
(8)$$Q_{hw}=\rho_{a}\cdot C_{ap}\cdot q_{hw}\cdot (T_{hw}-T_{a})$$
where $q_{hw}(m^{3}/s)$ is the air flow rate when the air conditioner blows hot air; $T_{hw}$(°C) is the heat generated during the operation of the air conditioner.
(5)Heat balance equation for building the structure and ventilation system

When establishing the heat balance equation, it is essential to take into account the heat entering the plant factory through both the building structure and the ventilation system, as indicated in Equation (9).
(9)$$Q_{v}=c_{r}\cdot A_{b}\cdot (T_{b}-T_{a})+\rho_{a}\cdot C_{ap}\cdot q_{f}\cdot (T_{out}-T_{a})$$
where $c_{r}$ is the heat transfer coefficient of the wall; $\;A_{b}(m^{2})$ is the internal wall area of the plant factory; $T_{b}$(°C) is the temperature of the PVC panels; $q_{f}(m^{3}/s)$ is the air flow rate of the outside air entering the inside of the plant factory; $T_{out}$ is the outside temperature.
(6)Crop transpiration model

This paper employs the method described in reference [34] to calculate the amount of heat released through crop transpiration, represented by Equation (10):(10)$$Q_{plant}=A_{P}\lambda E$$
where λ(kJ/g) is the latent heat of water evaporation; $E(g/m^{2}s)$ is the evaporation rate. According to references [34,35], represented by Equation (11):(11)$$\lambda E=R_{n}\frac{\left(1+e^{-KLAI}\right)}{1+\beta }$$
where ${\;R}_{n}(w/m^{2})$ is the net adiation intensity of crop canopy; K is the decay coefficient of LED lamps; β is the Bowen ratio. By substituting Equation (11) into Equation (10), the heat released from crop transpiration can be obtained, as represented by Equation (12):(12)$$Q_{plant}=A_{P}\lambda E=A_{P}R_{n}\frac{\left(1+e^{-KLAI}\right)}{1+\beta }$$

Combined with the experimental conditions in this study, β can be expressed by Equation (13):(13)$$\beta =\gamma \frac{T_{p}-T_{a}}{e_{0}RH_{in}(exp\left(17.4T_{p}/239+T_{p}\right)-exp\left(17.4T_{p}/239+y_{1}\right))}$$
where γ(kPa/°C) is the thermometer constant; $\;e_{0}(kPa)$ is the saturated water vapor pressure of the air in the factory at 0 °C; $RH_{in}$ is the relative humidity inside the plant factory.
(7)Heat balance equation for conveying cold air

This article mainly uses air conditioning to achieve indoor temperature decrease. While delivering cool air into the plant factory, the air conditioning controls the ventilation rate to achieve the desired temperature regulation. The energy balance equation for conveying cold air is established, which is represented by Equation (14).
(14)$$Q_{ca}=\rho_{a}\cdot C_{ap}\cdot q_{ca}\cdot (T_{ca}-T_{a})$$
where $q_{ca}(m^{3}/s)$ is the air flow rate when the air conditioner blows cold air; $T_{ca}$(°C) is the heat generated during the operation of the air conditioner.
(8)Humidity balance equation for crops

The humidity brought into the plant factory by the crops [32] through transpiration is shown in Equation (15):(15)$$W_{crops}=A_{P}\cdot E$$
(9)Humidity balance equation for humidifier

The increase in humidity in the plant factory is mainly affected by crop transpiration and a humidifier. The effect of soil and nutrient solution evaporation can be ignored compared to the effect of crop transpiration and a humidifier. The formula for crop transpiration has been given in the previous text. The humidity content of the air supplied by the humidifier is given by Equation (16).
(16)$$W_{hum}=\rho_{a}\cdot q_{hum}\cdot (H_{hum}-H_{a})$$
where $H_{hum}(kg/kg)$ is the humidity ratio of the supply air of the humidifier.

According to thermodynamics, the relationship between the humidity ratio and relative humidity in a plant factory is given by Equation (17):(17)$$H_{a}=\frac{622RH_{in}P_{0}}{P-RH_{in}P_{0}}$$
where $P(P_{a})$ is standard atmospheric pressure; $P_{0}(P_{a})$ is the water vapor partial pressure at different temperatures. Each temperature has a corresponding value for $P_{0}$.
(10)Humidity balance equation of dehumidifier

This paper mainly uses a dehumidifier to reduce the humidity in the plant factory. The humidity balance equation for the dehumidifier is established and represented by Equation (18).
(18)$$W_{dh}=\rho_{a}\cdot q_{dh}\cdot (H_{dh}-H_{a})$$
where $H_{dh}(kg/kg)$ is the humidity ratio of the supply air of the dehumidifier.
(11)Humidity balance equation for air-conditioning

The delivery of cold or hot air by the air conditioning system to the indoor environment has varying effects on the humidity content of the indoor air. The balance equations for two different scenarios are given by Equations (19) and (20).
(19)$$W_{ca}=\rho_{a}\cdot q_{ca}\cdot (H_{ca}-H_{a})$$
(20)$$W_{hw}=\rho_{a}\cdot q_{hw}\cdot (H_{hw}-H_{a})$$
where $H_{ca}(kg/kg)$ represents the humidity ratio of the cold air supplied by the air conditioning system; $H_{hw}(kg/kg)$ represents the humidity ratio of the hot air supplied by the air conditioning system.
(12)Humidity balance equation for the new air ventilation system

The exchange of air between the plant factory and the outside during operation is a disturbance factor that can potentially impact the stable operation of the plant factory environment. The formula for indoor humidity disturbance is given by Equation (21):(21)$$W_{v}=\rho_{a}\cdot q_{f}\cdot (H_{out}-H_{a})$$
where $H_{out}(kg/kg)$ is the humidity ratio of the outside air.

Define
$$T_{a}=y_{1},\;H_{a}=y_{2},\;q_{hw}=u_{1},\;q_{ca}=u_{2},\;q_{hum}=u_{3},\;q_{dh}=u_{4}.$$

The plant factory temperature model, represented by Equation (22), can be obtained by substituting Equations (3), (5)–(9), (12) and (14) into Equation (1). The plant factory humidity model, represented by Equation (23), can be obtained by substituting Equations (11), (16), (19)–(21) into Equation (2). The parameter meanings, values and unit in the equation can be found in Table 1.
(22)$$\begin{array}{c} \dot{y_{1}}=[A_{P}R_{n}(1-e^{-KLAI})/\rho_{a}VC_{ap}]/\{1+[\gamma (T_{p}-y_{1})]/[e_{0}Py_{2}/(622P_{0}+\\ P_{0}y_{2})(exp\left(17.4T_{p}/239+T_{p}\right)-exp\left(17.4T_{p}/239+y_{1}\right))]\}-[(u_{1}+u_{2}+u_{3}+\\ u_{4}+q_{f})/V+A_{P}/(r_{a}V)+(c_{r}A_{b})/(\rho_{a}VC_{ap})]y_{1}+(c_{r}A_{b})/(T_{hw}u_{1}+T_{ca}u_{2}+\\ T_{hum}u_{3}+T_{dh}u_{4})/V+k_{1}k_{2}\sum_{i=1}^{n}P_{i}t_{w}+(c_{r}A_{b}t_{b})/(\rho_{a}VC_{ap})+(A_{P}T_{P}+\\ q_{f}T_{out})/V \end{array}$$
(23)$$\begin{array}{c} \dot{y_{2}}=[{A_{P}R}_{n}(1-e^{-KLAI})/(\rho_{a}\lambda V)]/\{1+[\gamma (t_{p}-y_{1})]/[e_{0}Py_{2}/(622P_{0}+P_{0}y_{2})\\ (exp\left(17.4T_{p}/239+T_{p}\right)-exp\left(17.4T_{p}/239+y_{1}\right))]\}-((u_{1}+u_{2}+u_{3}+u_{4}+\\ q_{f})/V)y_{2}+(H_{hw}u_{1}+H_{ca}u_{2}+H_{hum}u_{3}+H_{dh}u_{4}+q_{f}H_{out})/V \end{array}$$

## 3. Nonlinear Adaptive Decoupling Control Based on Switching Mechanism

### 3.1. Nonlinear Decoupling Control Strategy 

Currently, the majority of plant factories utilize conventional controllers, such as PID controllers, to regulate the temperature and humidity levels within the environment. However, these traditional methods cannot handle the strong nonlinearity, strong coupling and parameter uncertainty in the environmental system. This paper proposes a multi-model-based nonlinear adaptive decoupling control method combining with a high-order neural network to accurately control the environmental factors in plant factories. Using the similar approach [28], the system of plant factory can be described as shown in Equation (24):(24)$$y\left(k+1\right)=-{\bar{A}}_{i}\left(z^{-1}\right)y\left(k\right)+B_{i}\left(z^{-1}\right)u\left(k\right)+v_{i}\left[x\left(k\right)\right]\;\;\;\;\;\;\;\;\;\;(i=1,2\ldots m)$$
(25)$$y\left(k\right)={\left[y_{1}\left(k\right),\ldots y_{n}(k)\right]}^{T}\in R^{n}$$
(26)$$u(k)={\left[u_{1}\left(k\right),\ldots u_{n}(k)\right]}^{T}\in R^{n}$$
where $y\left(k\right)$ and u(k) are *n*-dimensional input and output vectors; the quantity of known working points is represented by m. Make
(27)$$A_{i}(z^{-1})=I+z^{-1}{\bar{A}}_{i}\left(z^{-1}\right)$$
(28)$$B_{i}(z^{-1})={\bar{B}}_{i}(z^{-1})+{\overset{̿}{B}}_{i}(z^{-1})$$
${\bar{B}}_{i}(z^{-1})$ is a diagonal polynomial matrix with respect to $z^{-1}$, ${\overset{̿}{B}}_{i}(z^{-1})$ is a polynomial matrix with diagonal elements equal to zero with respect to $z^{-1}$. ${\bar{A}}_{i}(z^{-1})$ and ${\bar{B}}_{i}(z^{-1})$ are *n × n* dimensional polynomials with orders $n_{a}$ and $n_{b}$ with respect to $z^{-1}$. Then, $A_{i}(z^{-1})$ and $B_{i}(z^{-1})$ are expressed in the form of Equations (29) and (30):(29)$$A_{i}\left(z^{-1}\right)=I+A_{i1}z^{-1}+\cdots +A_{i\rho }z^{-\rho }+\cdots +A_{{in}_{a}}z^{-n_{a}}$$
(30)$$B_{i}(z^{-1})=B_{i0}+B_{i1}z^{-1}+\cdots +B_{i\rho }z^{-\rho }+\cdots +B_{{in}_{b}}z^{-n_{b}}$$
where I is the identity matrix; $A_{i1}\ldots A_{in_{a}}$ and $B_{i1}\ldots B_{in_{b}}$ are coefficient matrices; $A_{i\rho }$ and $B_{i\rho }$ are the coefficient of $z^{-\rho }$.
(31)$$v_{i}[x(k)]={\left[v_{i1}(x(k)),v_{i2}(x(k)),{\ldots v}_{in}(x(k))\right]}^{T}\in R^{n}$$
(32)$$v_{i}[x(k)]=A_{i}(z^{-1})y(k+1)-B_{i}(z^{-1})u(k)$$
(33)$$x(k)=[y^{T}(k),\ldots ,y^{T}(k-n_{s}+1),u^{T}(k),\ldots ,y^{T}(k-m_{s})]$$
where $v_{i}[x(k)]$ is an unknown continuously differentiable vector-valued nonlinear function that can be represented by Equations (31) and (32), and $∥v_{i}\left[x\left(k\right)\right]∥\leq △$; △ is a known positive real number. x(k) represents the input–output data vector of the system. Therefore, Equation (30) can be rewritten in the form shown in Equation (34):(34)$$y\left(k+1\right)=-{\bar{A}}_{i}\left(z^{-1}\right)y\left(k\right)+{\bar{B}}_{i}(z^{-1})u(k)+{\overset{̿}{B}}_{i}(z^{-1})u(k)+v_{i}[x(k)]$$

For the multivariate nonlinear control equation described by Equation (24) at the *i*-th operating point, we propose a nonlinear decoupling control strategy that integrates a feedback controller, a decoupling compensator and a nonlinear compensator, as shown in Figure 4. Here, ${\bar{H}}_{i}(z^{-1})$, ${\bar{R}}_{i}(z^{-1})$ and ${\bar{G}}_{i}(z^{-1})$ are the feedback controller parts of the system. These are diagonal polynomial matrices with respect to $z^{-1}$, which are used to control the system output y(k) to track the reference input w(k). The decoupling compensator
${\overset{=}{H}}_{i}(z^{-1})$ is a polynomial matrix with zero diagonal elements. It is used to eliminate the effect of coupling terms in the linear model. The nonlinear compensator ${\bar{K}}_{i}(z^{-1})$ is a diagonal polynomial matrix with respect to $z^{-1}$, which is used to remove the effect of nonlinear terms on the closed-loop system.

As shown in Figure 4, the control variable u(k) can be represented by Equation (35) as follows:(35)$$u(k)={{\bar{H}}_{i}}^{-1}(z^{-1})\{{\bar{R}}_{i}(z^{-1})w(k)-{\bar{G}}_{i}(z^{-1})y(k)-{\overset{̿}{H}}_{i}(z^{-1})u(k)-\;{\bar{K}}_{i}(z^{-1})v_{i}[x(k)]\}$$

By substituting the control variable u(k) into Equation (34), we obtain the equation for the closed-loop system, as depicted in Equation (36).
(36)$$[{\bar{H}}_{i}(z^{-1})A_{i}(z^{-1})+z^{-1}{\bar{B}}_{i}(z^{-1}){\bar{G}}_{i}(z^{-1})]y(k+1)={\bar{B}}_{i}(z^{-1})\bar{R}(z^{-1})w(k)+\;\left[{\bar{H}}_{i}(z^{-1}){\overset{̿}{B}}_{i}(z^{-1})-{\bar{B}}_{i}(z^{-1}){\overset{̿}{H}}_{i}(z^{-1})]u(k)+[{\bar{H}}_{i}(z^{-1})-{\bar{B}}_{i}(z^{-1}){\bar{K}}_{i}(z^{-1})]v_{i}[x(k)\right]$$
where $\left[{\bar{H}}_{i}(z^{-1})A_{i}(z^{-1})+z^{-1}{\bar{B}}_{i}(z^{-1}){\bar{G}}_{i}(z^{-1})\right]$, ${\bar{B}}_{i}(z^{-1}){\bar{R}}_{i}(z^{-1})$ and $\left[{\bar{H}}_{i}(z^{-1})-{\bar{B}}_{i}(z^{-1}){\bar{K}}_{i}(z^{-1})\right]$ are all polynomial matrices; $\left[{\bar{H}}_{i}(z^{-1}){\overset{̿}{B}}_{i}(z^{-1})-{\bar{B}}_{i}(z^{-1}){\overset{̿}{H}}_{i}(z^{-1})\right]$ is a polynomial matrix with zero diagonal elements. By choosing appropriate ${\bar{H}}_{i}(z^{-1})$, ${\bar{R}}_{i}(z^{-1})$ and ${\bar{G}}_{i}(z^{-1})$, it is possible to achieve system output y(k) that tracks reference input w(k) and eliminate steady-state errors as much as possible. By selecting the appropriate ${\overset{̿}{H}}_{i}(z^{-1})$, it is possible to eliminate $\left[{\bar{H}}_{i}(z^{-1}){\overset{̿}{B}}_{i}(z^{-1})-{\bar{B}}_{i}(z^{-1}){\overset{̿}{H}}_{i}(z^{-1})\right]$ influence on the system and minimize the impact between different loops. By selecting the appropriate ${\bar{K}}_{i}(z^{-1})$, the influence of nonlinearity caused by $v_{i}[x(k)]$ on the system can be minimized by $\left[{\bar{H}}_{i}(z^{-1})-{\bar{B}}_{i}(z^{-1}){\bar{K}}_{i}(z^{-1})]v_{i}[x(k)\right]$.

To provide a more intuitive representation of the system control method, the control variable u(k) can also be written in the form shown in Equation (37):(37)$$u_{\rho }(k)=\frac{1}{h_{\rho \rho }(z^{-1})}\{r_{ip}(z^{-1})w_{\rho }(k)-g_{ip}(z^{-1})y_{\rho }(k)-[h_{\rho 1}^{i}(z^{-1})u_{\rho }(k)+\cdots +\;h_{\rho ,\rho -1}^{i}(z^{-1})u_{\rho -1}(k)+h_{\rho ,\rho +1}^{i}(z^{-1})u_{\rho +1}(k)+\cdots +h_{\rho n}^{i}(z^{-1})u_{n}(k)]\;-k_{i\rho }(z^{-1})v_{i\rho }[x(k)]\}\;\;\;\;\;\;\;\;\;\;\;\;\;\;\;\;\;\;\;\;\;\;\;\;\;\;\;\;\;\;\;\;\;\;\;\;\;\;\;\;\;\;\;\;\;\;\;\;\;\;\;\;\;\;\;\;\;\;\;\;\;\;\;\;\;(\rho =1,2\ldots n)$$
where the diagonal elements of ${\bar{R}}_{i}(z^{-1})$, ${\bar{G}}_{i}(z^{-1})$ and ${\bar{K}}_{i}(z^{-1})$, denoted as $r_{i\rho }(z^{-1})$, $g_{i\rho }(z^{-1})$ and $k_{i\rho }\left(z^{-1}\right)$, respectively, correspond to the *i*-th diagonal element. $h_{\rho ,\varsigma }^{i}(z^{-1})$ represents the ρ-th row and ς-th column element of the polynomial matrix $H_{i}(z^{-1})$.
(38)$$H_{i}(z^{-1})={\bar{H}}_{i}(z^{-1})+{\overset{̿}{H}}_{i}(z^{-1})$$

From Equation (37), it can be seen that the control variable $u_{\rho }(k)$ of the ρ-th control loop is only related to the reference input $w_{\rho }(k)$, the output $y_{\rho }(k)$, the control variables $u_{1}(k),u_{2}(k),\ldots u_{\rho -1}(k),u_{\rho +1}(k+1),\ldots u_{n}(k)$ of other loops, as well as the nonlinear term $v_{i\rho }[x(k)]$.

### 3.2. Parameters Selection 

Due to the complex characteristics of the controlled object, a minimum variance control method is proposed to design the controller in this paper. The performance index shown in Equation (39) is introduced, which is similar to the one in [36]:(39)$$J=∥P_{i}\left(z^{-1}\right)y\left(k+1\right)-R_{i}\left(z^{-1}\right)w\left(k\right)+Q_{i}\left(z^{-1}\right)u\left(k\right)+\;S_{i}\left(z^{-1}\right)u\left(k\right){+K_{i}\left(z^{-1}\right)v_{i}\left[x\left(k\right)\right]∥}^{2}$$
where w(k) is a reference vector; $R_{i}\left(z^{-1}\right)$, $Q_{i}\left(z^{-1}\right)$ and $K_{i}\left(z^{-1}\right)$ are diagonal polynomial matrices with respect to $z^{-1}$; $P_{i}(z^{-1})$ is a weighted polynomial and $S_{i}(z^{-1})$ is a non-diagonal polynomial matrix with respect to $z^{-1}$. The Diophantine equation is introduced as shown in Equation (40).
(40)$$P_{i}(z^{-1})=F_{i}(z^{-1})A_{i}(z^{-1})+z^{-1}G_{i}(z^{-1})$$

Since matrix multiplication is required when solving the Diophantine equation for multivariable systems, the concept of pseudo-commutation matrix [37] is introduced. There exist unique solutions, $F_{i}(z^{-1})$ and $G_{i}(z^{-1})$, that minimize the performance index in Equation (39). Therefore, the optimal control rate can be represented by Equation (41):(41)$$[F_{i}(z^{-1}){\bar{B}}_{i}(z^{-1})+Q_{i}(z^{-1})]u(k)=R_{i}(z^{-1})w(k)-G_{i}(z^{-1})y(k)-\;\left[F_{i}(z^{-1}){\overset{̿}{B}}_{i}(z^{-1})+S_{i}(z^{-1})]u(k)-[F_{i}(z^{-1})+K_{i}(z^{-1})]v_{i}[x(k)\right]$$

By left-multiplying Equation (41) with ${\bar{B}}_{i}(z^{-1})$ and left-multiplying Equation (34) with $\left[F_{i}(z^{-1}){\bar{B}}_{i}(z^{-1})+Q_{i}(z^{-1})\right]$, we obtain Equation (42):(42)$${[P_{i}(z^{-1}){\bar{B}}_{i}(z^{-1})+Q_{i}(z^{-1})A}_{i}(z^{-1})]y(k+1)={\bar{B}}_{i}(z^{-1})R_{i}(z^{-1})w(k)+\;[Q_{i}\left(z^{-1}\right){\overset{̿}{B}}_{i}\left(z^{-1}\right)-{\bar{B}}_{i}(z^{-1})S_{i}(z^{-1})]u(k)+[Q_{i}\left(z^{-1}\right)-\;{\bar{B}}_{i}(z^{-1})K_{i}(z^{-1})]v_{i}[x(k)]$$

By left-multiplying Equation (34) with $G_{i}(z^{-1})$ and left-multiplying Equation (41) with $A_{i}(z^{-1})$, we obtain the closed-loop system equation as shown in Equation (43):(43)$$\left\{P_{i}(z^{-1})B_{i}(z^{-1})+A_{i}(z^{-1})[Q_{i}(z^{-1})+S_{i}(z^{-1})]\}u(k)=A_{i}(z^{-1})R_{i}(z^{-1}\right)\;w(k)+[P_{i}(z^{-1})+A_{i}(z^{-1})K_{i}(z^{-1})]v_{i}[x(k)]$$

In order to eliminate steady-state errors and achieve static decoupling of the system, the following equations, as shown in Equations (44)–(46), should be satisfied when selecting the weighting matrices $P_{i}(z^{-1})$, $Q_{i}(z^{-1})$, $R_{i}(z^{-1})$, $K_{i}(z^{-1})$ and $S_{i}(z^{-1})$.
(44)$${P_{i}(1){\bar{B}}_{i}(1)+Q_{i}(1)A}_{i}(1)={\bar{B}}_{i}(1)R_{i}(1)$$
(45)$$Q_{i}\left(1\right){\overset{̿}{B}}_{i}\left(1\right)={\bar{B}}_{i}(1)S_{i}(1)$$
(46)$$Q_{i}\left(1\right)={\bar{B}}_{i}(1)K_{i}(1)$$

Additionally, in order to ensure stability of the closed-loop system, it is also necessary to meet the condition specified in Equation (47).
(47)$$det\{P_{i}(z^{-1})B_{i}(z^{-1})+A_{i}(z^{-1})[Q_{i}(z^{-1})+S_{i}(z^{-1})]\}\neq 0,|z|\geq 1$$

Define
$${\bar{H}}_{i}\left(z^{-1}\right)=F_{i}\left(z^{-1}\right){\bar{B}}_{i}\left(z^{-1}\right)+Q_{i}\left(z^{-1}\right),\;{\bar{R}}_{i}\left(z^{-1}\right)=R_{i}\left(z^{-1}\right),{\bar{G}}_{i}\left(z^{-1}\right)=G_{i}(z^{-1})\;{\overset{̿}{H}}_{i}\left(z^{-1}\right)=F_{i}(z^{-1}){\overset{̿}{B}}_{i}(z^{-1})+S_{i}(z^{-1}),{\bar{K}}_{i}\left(z^{-1}\right)=F_{i}(z^{-1})+K_{i}(z^{-1})$$

The nonlinear decoupling controller can be represented by Equation (48):(48)$${\bar{H}}_{i}\left(z^{-1}\right)u(k)={\bar{G}}_{i}(z^{-1})w(k)-{\bar{G}}_{i}(z^{-1})y(k)-{\overset{̿}{H}}_{i}(z^{-1})u(k)-{\bar{K}}_{i}(z^{-1})v_{i}[x(k)]\}$$

A linear decoupling controller can be obtained if the nonlinear terms are not taken into consideration, which is represented by Equation (49):(49)$${\bar{H}}_{i}\left(z^{-1}\right)u(k)={\bar{G}}_{i}(z^{-1})w(k)-{\bar{G}}_{i}(z^{-1})y(k)-{\overset{̿}{H}}_{i}(z^{-1})u(k)$$

### 3.3. Adaptive Control Algorithm 

In the equation described by Equation (34), if $A_{i}(z^{-1})$ and $B_{i}(z^{-1})$ are both known, and the nonlinear part $v_{i}[x\left(k\right)]$ has a relatively small impact on the system, using the generalized minimum variance control method can complete the tracking of set points. However, during the actual operation of the plant factory, the parameters will vary with the material properties. This requires the parameters of the controller to also change accordingly. Therefore, this paper uses the following identification algorithm to identify the system parameters. Let $M_{1i}$ denote the linear estimation model of the system, represented as Equation (50):(50)$${\hat{y}}_{1}(k+1)={\hat{\theta }}_{1i}^{T}(k)\phi (k)$$
(51)$$\phi (k)={\left[y(k),\ldots ,y(k-n_{s}+1),u(k),\ldots ,u(k-m_{s})\right]}^{T}$$
where ϕ(k) is the input–output data vector of the system; ${\hat{\theta }}_{1i}(k)$ is the estimated value of the parameters θ of the system obtained through parameter identification at time k. In this paper, the projection algorithm with dead zone as shown in Equations (52) and (53) is employed to identify the parameters in each sampling period.
(52)$${\hat{\theta }}_{1i}(k)={\hat{\theta }}_{1i}(k-1)+\frac{\mu_{1}(k)\phi (k-1)e_{1i}^{T}(k)}{1+\phi^{T}(k-1)\phi (k-1)}$$
(53)$$\mu_{1}(k)=\{\begin{array}{cc} 1 & \;\mathrm{if}\;∥e_{1}(k)∥\;>\;4\Delta \\ 0 & \;\mathrm{else}\; \end{array}$$

The error value of the linear model in the equation can be represented by Equation (54): (54)$$e_{1}(k)=y(k)-{\hat{y}}_{1}(k)=y(k)-{\hat{\theta }}_{1i}^{T}(k)\phi (k)$$

Suppose that, at time k, the coefficient matrices $A_{i}(z^{-1})$ and $B_{i}(z^{-1})$ of the system are estimated based on the estimated model Equation (50) and the identification algorithm Equation (52), respectively, and the estimates are $A_{i}^{h}(z^{-1})$ and $B_{i}^{h}(z^{-1})$. By calculating the corresponding $P_{i}(z^{-1})$, $Q_{i}(z^{-1})$, $R_{i}(z^{-1})$ and then substituting these estimates into Equation (49), the linear adaptive decoupling controller $C_{11}$ can be obtained by Equation (55):(55)$${\bar{H}}_{i}\left(z^{-1}\right)u(k)={{\bar{G}}_{i}}^{h}(z^{-1})w(k)-{{\bar{G}}_{i}}^{h}(z^{-1})y(k)-{{\overset{̿}{H}}_{i}}^{h}(z^{-1})u(k)$$

For weak nonlinear systems, using linear adaptive controllers can achieve control requirements. However, in the case of the microclimate environment of a plant factory exhibiting strong nonlinearity, the numerical values of the nonlinear component $v_{i}[x\left(k\right)]$ tend to be large. In such circumstances, relying solely on linear system controllers becomes challenging to meet the control requirements. Therefore, neural network estimates are used to estimate nonlinear terms, and linear terms are combined to approximate the original control system. The system’s nonlinear estimation model $M_{2i}$ can be represented by Equation (56):(56)$${\hat{y}}_{2}\left(k+1\right)={\hat{\theta }}_{2i}^{T}\left(k\right)\phi \left(k\right)+{\hat{v}}_{i}[x\left(k\right)]$$

Since the nonlinear part $v_{i}[x\left(k\right)]$ in y(k) is an uncertain term within the system, a high-order neural network estimation is employed to handle it. The algorithm shown in Equations (57) and (58) is utilized to identify the parameter θ.
(57)$${\hat{\theta }}_{2i}(k)={\hat{\theta }}_{2i}(k-1)+\frac{\mu_{2}(k)\phi (k-1)e_{2i}^{T}(k)}{1+\phi^{T}(k-1)\phi (k-1)}$$
(58)$$\mu_{2}(k)=\{\begin{array}{cc} 1 & \;\mathrm{if}\;∥e_{2}(k)∥>4\Delta \\ 0 & \;\mathrm{else}\; \end{array}$$

The calculation error of the nonlinear model in the equation can be represented by Equation (59):(59)$$e_{2}\left(k\right)=y\left(k\right)-{\hat{y}}_{2}\left(k\right)=y\left(k\right)-{\hat{\theta }}_{2i}^{T}\left(k\right)\phi \left(k\right)-{\hat{v}}_{i}[x\left(k\right)]$$

Based on the high-order neural network estimation of the nonlinear term $v_{i}[x\left(k\right)]$, the nonlinear adaptive decoupling controller $C_{21}$ can be obtained by Equation (60). The structure diagram of the adaptive control algorithm is shown in Figure 5.
(60)$${\bar{H}}_{i}\left(z^{-1}\right)u\left(k\right)={{\bar{G}}_{i}}^{l}\left(z^{-1}\right)w\left(k\right)-{{\bar{G}}_{i}}^{l}\left(z^{-1}\right)y\left(k\right)-{{\overset{̿}{H}}_{i}}^{l}\left(z^{-1}\right)u\left(k\right)-\;{{\bar{K}}_{i}}^{l}(z^{-1}){\hat{v}}_{i}[x(k)]$$

### 3.4. Switching Control

A linear model obtained through the identification algorithm is used for designing a linear controller. Nevertheless, such linear controllers frequently fall short of achieving desired outcomes in practical microclimate environments. Conversely, relying exclusively on a nonlinear model to describe the system and design a corresponding nonlinear controller poses challenges in maintaining system stability throughout its operation. In fact, it is often difficult to determine whether the characteristics of the plant factory environment are linear or nonlinear at the current moment. Therefore, inspired by the literature [38], the switching control as shown in Figure 6 is introduced to ensure the stable operation of the system and to provide better control performance. The switching control strategy adopted in this paper needs to switch between the generalized minimum variance linear controller and the nonlinear decoupling controller. The generalized minimum variance linear controller is used to ensure the stable operation of the system, and the nonlinear decoupling controller compensates for the nonlinear part in the model, reducing the impact of unmolded nonlinear parts on the output and achieving an excellent control performance of the system.

The linear model contained in the switching mechanism in the figure is represented by Equation (61):(61)$${\hat{y}}_{1}\left(k+1\right)={\bar{A}}_{i}\left(z^{-1}\right)y\left(k\right)+{\bar{B}}_{i}(z^{-1})u(k)+{\overset{̿}{B}}_{i}(z^{-1})u(k)$$

The nonlinear model contained in the switching control in the figure is represented by Equation (62): (62)$${\hat{y}}_{2}\left(k+1\right)={\bar{A}}_{i}\left(z^{-1}\right)y\left(k\right)+{\bar{B}}_{i}\left(z^{-1}\right)u\left(k\right)+{\overset{̿}{B}}_{i}\left(z^{-1}\right)u\left(k\right)+{\hat{v}}_{i}[x\left(k\right)]$$

Selecting a suitable switching function allows the system to select the optimal control method based on the current system characteristics at time k. The switching mechanism is represented by Equations (63)–(65): (63)$$J_{i}(k)=\sum_{i=1}^{n}\frac{\mu_{i}(l)[e_{i}^{2}(l)-16\Delta^{2}]}{1+\phi {\left(l-1\right)}^{T}\phi (l-1)}+\alpha \sum_{l=k-N+1}^{k}\left[1-\mu_{i}(l)\right]e_{i}^{2}(l)$$
(64)$$\mu_{i}(k)=\{\begin{array}{cc} 1 & \;\mathrm{if}\;\left|e_{i}(k)\right|>4\Delta \\ 0 & \;\mathrm{else}\; \end{array}$$
(65)$$e_{i}\left(k\right)=y\left(k\right)-{\hat{y}}_{i}\left(k\right)\;\;\;\;\;\;\;\;\;\;\;\;\;\;(i=1,2)$$

In the equation, N is a positive integer, α is a positive constant, $e_{1}\left(k\right)$ is the error value of the linear model and $e_{2}\left(k\right)$ is the error value of the nonlinear model. When the value of i is 1, a linear controller is used to predict the system output, and when the value of i is 2, a nonlinear controller is used to estimate the system output. During this process, model parameters are automatically renewed by the identification algorithm mentioned above. When the system starts to operate, $J_{1}\left(k\right)$ and $J_{2}\left(k\right)$ are calculated in turn, and the controller corresponding to the minimum is selected to act on the controlled object.

### 3.5. High-Order Neural Network for Unmolded Dynamics 

High-order neural network (HONN) is a type of neural network that can approximate the given nonlinear function with arbitrary accuracy within a specified compact set. Compared with neural networks that only have summing units between layers and require increasing the number of intermediate nodes to solve nonlinear problems, HONN has fewer hidden layer nodes and fewer parameters to be adjusted when estimating non-limiting terms. Therefore, this article chooses HONN for the controller design. The structure of HONN can be represented by Equations (66)–(69):(66)$$G_{nn}\left(W,\bar{z}\right)=W^{T}\psi \left(\bar{z}\right),W\in R^{l\times n}\;\;\;\;\;\psi (\bar{z})\in R^{l}$$
(67)$$\psi (\bar{z})={\left[\psi_{1}(\bar{z}),\psi_{2}(\bar{z}),\ldots \psi_{l}(\bar{z})\right]}^{T}$$
(68)$$\bar{z}={\left[z_{1},z_{1},z_{1}\ldots z_{l}\right]}^{T}\in \Omega_{z}\subset R^{q}$$
(69)$$\psi_{i}\left(z\right)=\prod_{j\in I_{i}}{\left[\psi \left(z_{j}\right)\right]}^{\mu_{j}\left(i\right)}\;\;\;\;\;\;\;(i=1,2,3\ldots l)$$
where $G_{nn}(W,\bar{z})$ is the output of the neural network; W is the weight of the neural network; $\bar{z}$ is the input of the neural network; l is the number of hidden layer nodes of the neural network; q is the number of input values of the neural network; $\left\{I_{i}\right\}$ is the unordered subset of {1,2,3…q}; $\mu_{j}(i)$ is a non-negative integer; and $\psi (z_{j})$ is the S function.

For any nonlinear function, there exists an ideal weight matrix $W^{*}$ that yields the output of the neural network in the form depicted by Equation (70).
(70)$$G_{nn}(W,\bar{z})=W^{*T}\psi (\bar{z})+\zeta_{z}$$
where $\zeta_{z}$ is the estimation error of the neural network. The estimation error of the neural network can be reduced by gradually increasing the number of network nodes, and $\zeta_{z}$ can approach zero when l is sufficiently large. For all $\bar{z}\in \Omega_{z}\subset R^{q}$, there exists an ideal matrix represented by Equation (71) that minimizes $∥\zeta_{z}∥$, namely:(71)$$W^{*}=arg\underset{W\in \Omega_{w}}{min}\{\underset{z\in \Omega_{z}}{sup}|G(z(k))-W^{T}\psi (\bar{z}(k))|\}\;\;\;\;\;\;\Omega_{z}\in R^{q},\Omega_{w}\in R^{l\times q}$$

In this paper, the nonlinear term $v_{i}[x\left(k\right)]$ can be presented by Equation (72):(72)$$\begin{array}{c} {\hat{v}}_{i}\left[x\left(k\right)\right]=W^{*T}\psi \left(\bar{z}\right)+\zeta_{z}\\ ={arg\underset{W\in \Omega_{w}}{min}\{\underset{z\in \Omega_{z}}{sup}|G(z(k))-W^{T}\psi (\bar{z}(k))|\}}^{T}\psi \left(\bar{z}\right)+\zeta_{z}\;\;\;\;\Omega_{z}\in R^{q},\;\Omega_{w}\in R^{l\times q} \end{array}$$

## 4. Simulation Results

This paper focuses on the microclimate environment of a plant factory and takes sweet peppers as the experimental subject to carry out a set-point tracking experiment, parameter uncertainty experiment and multi-disturbance experiment. To validate the robustness and adaptability of the control algorithm proposed, a nonlinear adaptive decoupling controller is adopted to control the internal environment of the factory. Nonlinear adaptive decoupling controllers are used to control the internal environment of the plant factory to verify the robustness and adaptability of the proposed control algorithm.

Sweet peppers are widely cultivated worldwide due to their high nutritional and economic value. The principles for setting the temperature and humidity during the cultivation process in the plant factory are as follows.
(1)Significant temperature differences for the microclimate environment can promote thicker stems, denser leaves, increased leaf area, and improved absorption and utilization of light energy by plants. In general, higher temperatures during the daytime are beneficial for photosynthesis and nutrient absorption, promoting energy accumulation and plant growth. Lower temperatures at night help plants to respire, distribute nutrients, enhance root growth, boost metabolic activity and strengthen their disease resistance.(2)Crop humidity requirements vary at different growth stages. Higher humidity is needed during seed germination and seedling stages, and it can gradually be reduced as crops grow and develop.(3)From an energy utilization perspective, control systems should operate at a reasonable level that meets environmental demands. Excessive high or low temperatures, as well as excessive high or low humidity, can increase equipment load and operational costs.

In this paper, the ripening stage of sweet peppers is considered in the experiment. It is important to avoid excessively high or low temperatures. When the temperature drops below 13 °C or exceeds 32 °C, it is not conducive to crops development. The optimal range of humidity during the ripening stage for crops development is 60–70%. The temperature and humidity settings for simulating the growing environment of the crops throughout a 24 h period are presented in Table 2. The 24 h period was divided into 10 segments for conducting simulation experiments. Set-point tracking experiments are conducted from 06:00 a.m. to 20:00 p.m. Parameter uncertainty experiments and multi-disturbance experiments are implemented from 20:00 p.m. to 06:00 a.m. The experimental design is as follows.

From 06:00 a.m. to 07:00 a.m., the internal temperature of the factory is set to 20 °C, and the relative humidity is set to 65%. During this period, the air conditioner blows out warm air ($u_{1}$), and the humidifier ($u_{3}$) works. From 07:00 a.m. to 08:00 a.m., the temperature setting inside the factory remains unchanged, and the relative humidity is set to 60%. During this period, the air conditioner still blows out warm air ($u_{1}$), and the dehumidifier ($u_{4}$) works. From 08:00 a.m. to 12:00 a.m., the temperature set values are increased gradually from 26 °C to 30 °C, and the relative humidity is set to 70%. At this time, the air conditioner blows warm air ($u_{1}$), and the humidifier ($u_{4}$) works. From 12:00 a.m. to 16:00 p.m., the temperature set value drops from 28 °C to 23 °C, and the relative humidity drops from 70% to 65% and then returns to 70%. During this process, the air conditioner exhales cold air into the plant factory ($u_{2}$), and the dehumidifier ($u_{4}$) and humidifier ($u_{3}$) worked from 12:00 a.m. to 14:00 p.m. and 14:00 p.m. to 16:00 p.m., respectively. From 16:00 p.m. to 06:00 a.m., the relative humidity inside the plant factory is set to 60%, and the temperature gradually decreases from 23 °C to 15 °C. At this time, the air conditioner blows cold air into the plant factory ($u_{2}$), and the dehumidifier ($u_{4}$) works continuously.

After the system reaches steady operation, parameter uncertainty experiments are implemented. At 20:00, the heat transfer coefficient ($c_{r}$) of the PVC panels is changed to verify if the system can quickly regain stability. To assess the control performance of the system under multi-disturbances, the plant factory undergoes a simulation of ventilation operation where air from the outside is introduced into the facility from 00:00 a.m. to 06:00 a.m. The ventilation rate ($q_{f}$), outside temperature ($T_{out}$) and outside humidity ($H_{out}$) are set to random values within a certain range to simulate the multi-disturbances in actual situations. 

The parameters of the controller are designed as follows: After Euler transformation, the initial operating point is: $A_{11}(z)=(0.1038z-0.1038)/(z^{2}-1.9985z+0.9985),\;A_{12}(z)=(-0.002841z+0.002837)/(z^{2}-1.9985z$
+0.9985), $A_{21}(z)=(0.03792z-0.03792)/(z^{2}-1.9985z+0.9985),\;A_{22}(z)=(0.02841z-0.02837)/(z^{2}$
−1.9985z+0.9985). The order of the system is: $n_{a}=2$, $n_{b}=2$. The parameters of the nonlinear adaptive decoupling controller are set as follows: the number of hidden layers is 12. The parameters of the switching mechanism are α=1 and Δ=0.01.

For comparison, the traditional PID control method is adopted. The control effect of temperature tracking using the traditional PID control strategy is shown in Figure 7, and the control result of temperature using the traditional PID control strategy in parameter uncertainty experiment and multi-disturbance experiment is shown in Figure 8. The control effect of humidity tracking using the traditional PID control strategy is shown in Figure 9, and the control result of humidity using the traditional PID control strategy in parameter uncertainty experiment and multi-disturbance experiment is shown in Figure 10. Figure 11, Figure 12, Figure 13 and Figure 14 show the corresponding responses of control inputs using the traditional PID control strategy. The control effect of temperature tracking effect by nonlinear adaptive decoupling control strategy is shown in Figure 15, and the control result of temperature by nonlinear adaptive decoupling control strategy in parameter uncertainty experiment and multi-disturbance experiment is shown in Figure 16. The control effect of humidity tracking by nonlinear adaptive decoupling control strategy is shown in Figure 17, and the control result of humidity by nonlinear adaptive decoupling control strategy in parameter uncertainty experiment and multi-disturbance experiment is shown in Figure 18. Figure 19, Figure 20, Figure 21 and Figure 22 show the responses generated by the control inputs when using a nonlinear adaptive decoupling control strategy.

As observed from Figure 7, Figure 8, Figure 9 and Figure 10, it is evident that the traditional PID control exhibits considerable fluctuations, and the tracking effect is not good. Additionally, when the temperature and humidity change, the tracking time is relatively prolonged and sometimes there exists a steady-state error. Furthermore, the temperature has a substantial overshoot and takes a considerable amount of time to return to the steady state when the parameter changes. Moreover, the system shows larger fluctuations when subjected to multi-disturbances. 

From Figure 15, Figure 16, Figure 17 and Figure 18, it shows that by using the nonlinear adaptive decoupling control strategy proposed in this paper, the tracking time is significantly shortened, and the fluctuations in humidity are noticeably reduced when the temperature set point is changed. Similarly, when the humidity set point is adjusted, the system demonstrates the ability to promptly track the new set point, while minimizing the impact on temperature. This achieves a good decoupling effect. In the case of parameter uncertainty, the temperature and humidity values exhibit smaller overshoots and can quickly recover the set point during this period. When facing multi-disturbances, the temperature and humidity values fluctuate within a small range around the set point. Therefore, adopting the nonlinear adaptive decoupling control strategy can mitigate the effects of uncertainty on the system.

In a plant factory, the control system plays a vital role in regulating environmental parameters. Through control system optimization, timely activation of temperature and humidity regulation, devices can respond to the plant’s growth requirements and external environmental changes, while minimizing energy consumption. The control system self-adjusts to prevent unnecessary continuous operation once the system reaches the desired set point. Experimental results demonstrate that employing a high-order neural-network-based nonlinear adaptive decoupling control strategy reduces equipment working time compared to the traditional PID control method. The equipment can reach the set point more quickly. Moreover, under the influence of parameter uncertainties and multi-disturbance, the high-order neural-network-based nonlinear adaptive decoupling control strategy minimizes control variable fluctuations, mitigating the occurrence of excessively high or low temperatures and alleviating the equipment burden. In summary, employing a high-order neural-network-based nonlinear adaptive decoupling control strategy enables achieving desired control effects more efficiently, leading to reduced operational costs and equipment energy consumption.

To assess the control performance of both the traditional PID method and the proposed nonlinear adaptive decoupling control, the mean error and standard error are introduced as evaluation metrics. Table 3 shows the control results of the two controllers for the environmental temperature and humidity under different conditions. Through comparison, the proposed nonlinear adaptive decoupling control exhibits smaller numerical values in both evaluation metrics.

The feasibility and adaptability of the proposed control algorithms in a plant factory have been confirmed through experiments that track the set values of temperature and humidity, experiments with parameter uncertainties and experiments with multi-disturbances. Precise regulation of temperature and humidity in complex situations that may arise in a plant factory enables the provision of optimal growth conditions, meets different crop requirements for environmental temperature and humidity, promotes healthy growth and increases yields. This regulation also eliminates limitations imposed by natural environmental conditions on crop production. Additionally, it can optimize the production cycle of crops by controlling temperature and humidity variations at different stages. This allows for the acceleration or delay of plant growth and development, as well as the advancement or postponement of harvest time. This eliminates seasonal and geographical restrictions, resulting in more stable and sustainable agricultural production. Overall, a plant factory serves as essential components of smart agriculture and holds significant potential for sustainable agricultural development. They provide an efficient, resource-saving and high-quality agricultural production mode through the implementation of innovative technologies and environmental control methods.

## 5. Conclusions

This paper specifically investigates a plant factory environment system that is characterized by nonlinearity, strong coupling and multiple disturbances. Based on the energy balance, a nonlinear dynamic equation describing the temperature and humidity environment model of the plant factory is established, and corresponding control methods are introduced. A nonlinear adaptive decoupling control strategy using high-order neural network is proposed, which utilizes the powerful learning abilities of the high-order neural network to address nonlinear functions, so that the system can reduce nonlinearity impact on system. To meet the requirements of temperature and humidity regulation, a generalized minimum variance controller is designed. The mean error and standard error of the traditional controller for temperature control are 0.3615 and 0.8425, respectively. In contrast, the proposed control strategy has made significant improvements in both indicators, with results of 0.1655 and 0.6665, respectively. For humidity control, the mean error and standard error of the traditional controller are 0.1475 and 0.441, respectively. In comparison, the proposed control strategy has greatly improved on both indicators, with results of 0.0221 and 0.1541, respectively. The simulation results show that this control strategy can achieve tracking of the set value in a short time and has good robustness and adaptability. Simultaneously, in the case of parameter uncertainty and encountering multi-disturbances, the system exhibits less overshoot and only shows minor fluctuations. Therefore, the proposed method enables accurate regulation of microclimate environmental factors within a plant factory to fulfill the growth requirements of crops, consequently enhancing crop productivity and quality.

## Figures and Tables

**Figure 1 sensors-23-08323-f001:**
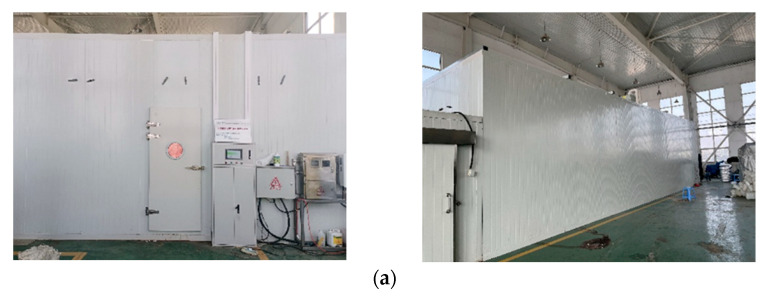
External architecture of the plant factory (**a**), and internal architecture diagram (**b**).

**Figure 2 sensors-23-08323-f002:**
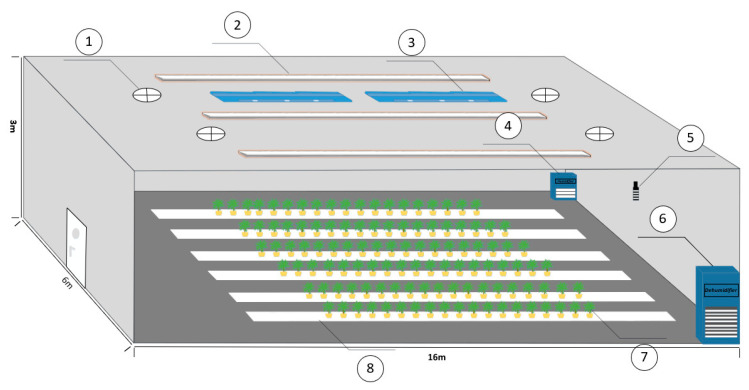
The internal environmental diagram of the plant factory. ①: Fresh air system ②: LED lamps ③: Air conditioning ④: Humidifier ⑤: Temperature and humidity sensor ⑥: Dehumidifier ⑦: Crops ⑧: Crop cultivation rack.

**Figure 3 sensors-23-08323-f003:**
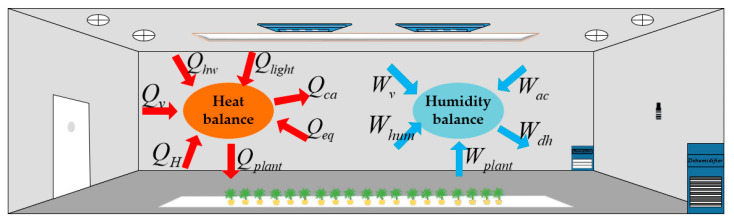
Schematic diagram of the heat balance and humidity balance of the plant factory.

**Figure 4 sensors-23-08323-f004:**
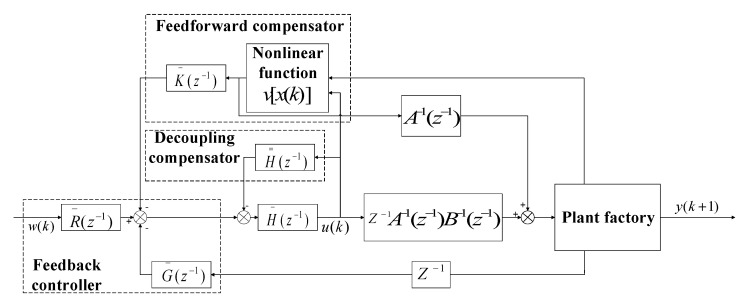
Schematic diagram of nonlinear decoupling control method.

**Figure 5 sensors-23-08323-f005:**
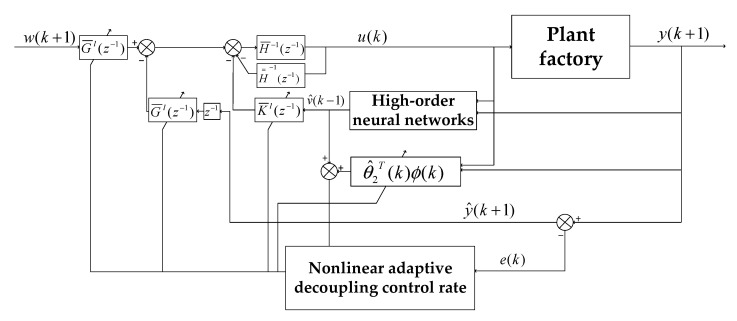
Schematic diagram of nonlinear adaptive decoupling control algorithm.

**Figure 6 sensors-23-08323-f006:**
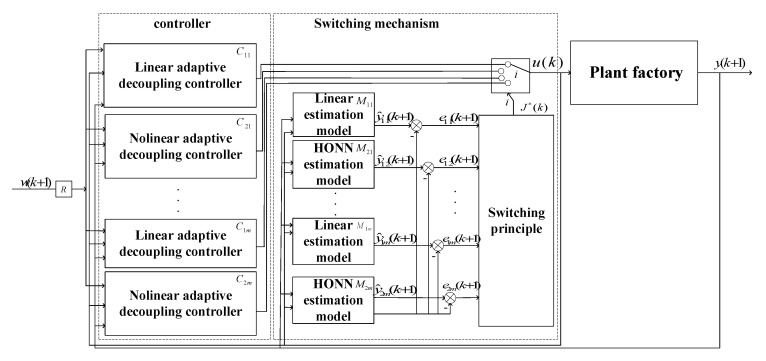
Schematic diagram of switching control based on multiple models.

**Figure 7 sensors-23-08323-f007:**
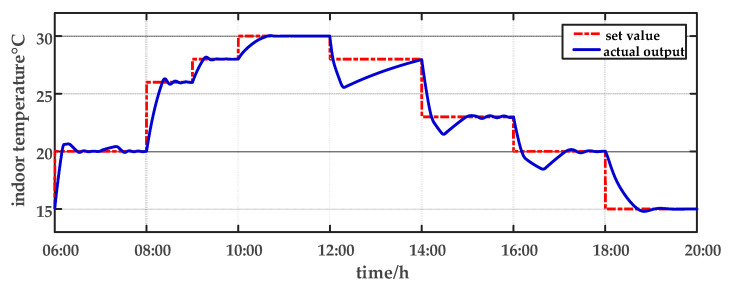
The temperature tracking effect using the traditional PID control strategy from 06:00 a.m. to 20:00 p.m.

**Figure 8 sensors-23-08323-f008:**
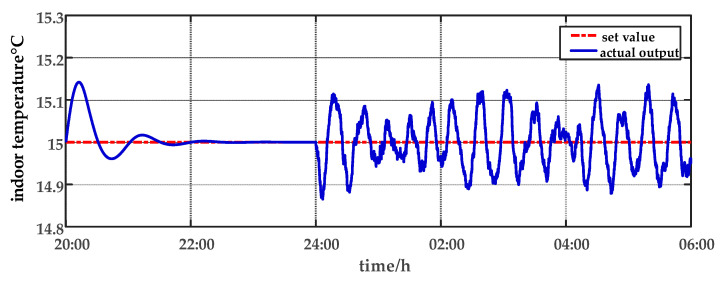
The effect of temperature using the traditional PID control strategy for parameter uncertainty experiment and multi-disturbance experiment from 20:00 p.m. to 06:00 a.m.

**Figure 9 sensors-23-08323-f009:**
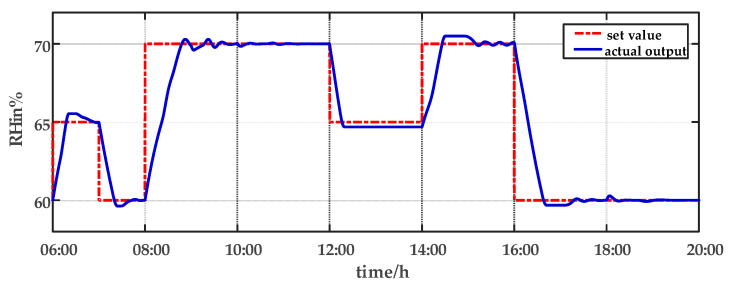
The humidity tracking effect using the traditional PID control strategy from 06:00 a.m. to 20:00 p.m.

**Figure 10 sensors-23-08323-f010:**
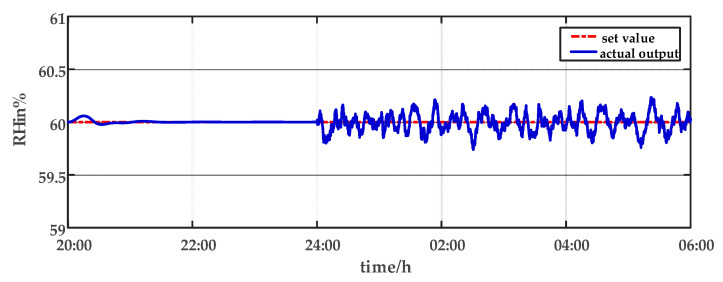
The effect of humidity using the traditional PID control strategy in parameter uncertainty experiment and multi-disturbance experiment from 20:00 p.m. to 06:00 a.m.

**Figure 11 sensors-23-08323-f011:**
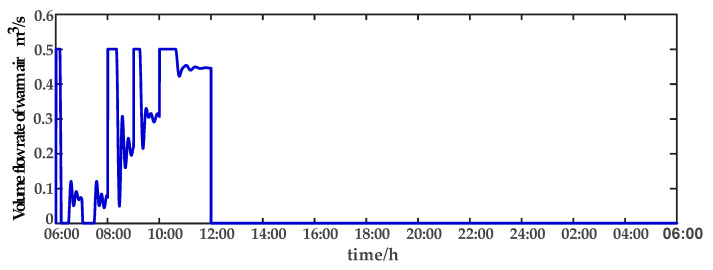
Volume flow rate of warm air ($u_{1}$) under the traditional PID control strategy.

**Figure 12 sensors-23-08323-f012:**
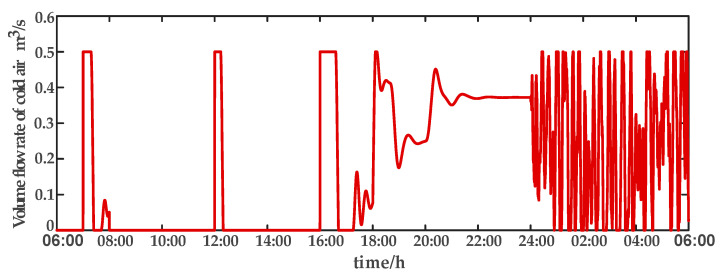
Volume flow rate of cold air ($u_{2}$) under the traditional PID control strategy.

**Figure 13 sensors-23-08323-f013:**
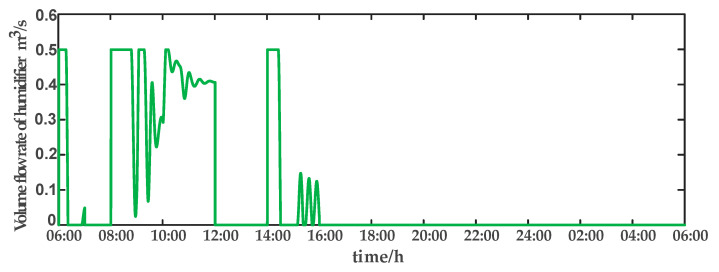
Volume flow rate of humidifier ($u_{3}$) under the traditional PID control strategy.

**Figure 14 sensors-23-08323-f014:**
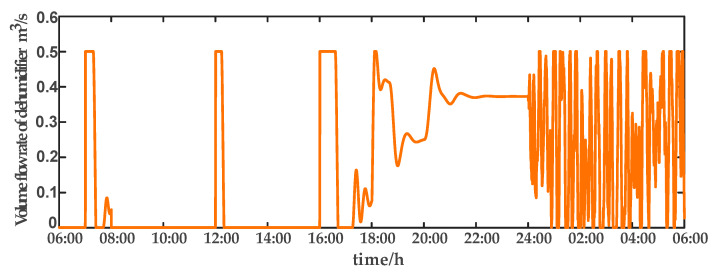
Volume flow rate of dehumidifier ($u_{4}$) under the traditional PID control strategy.

**Figure 15 sensors-23-08323-f015:**
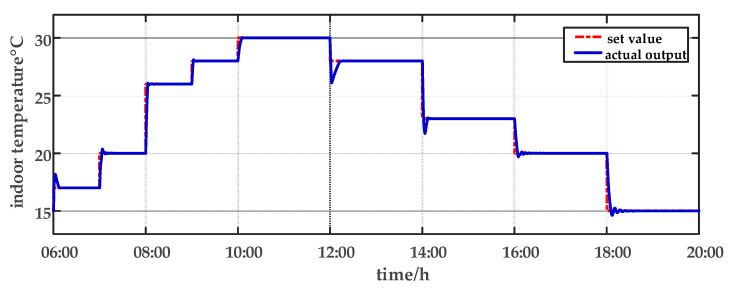
The temperature tracking effect by nonlinear adaptive decoupling control strategy from 06:00 a.m. to 20:00 p.m.

**Figure 16 sensors-23-08323-f016:**
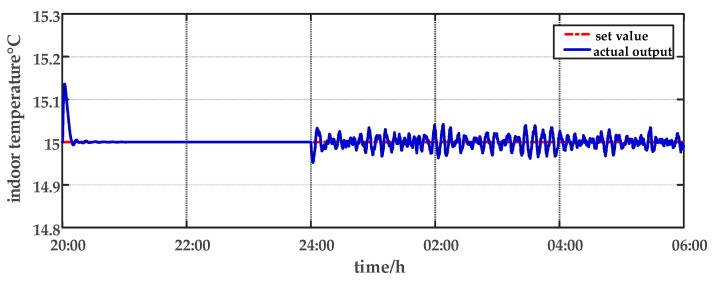
The effect of temperature by nonlinear adaptive decoupling control strategy in parameter uncertainty experiment and multi-disturbance experiment from 20:00 p.m. to 06:00 a.m.

**Figure 17 sensors-23-08323-f017:**
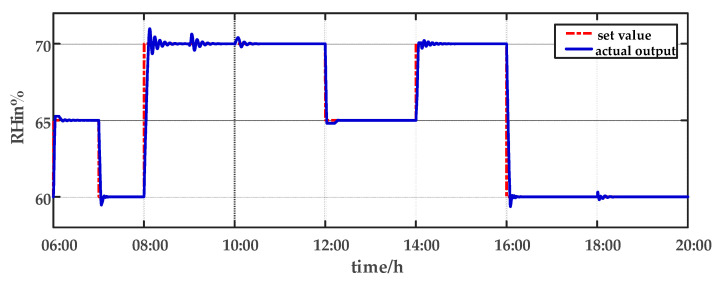
The humidity tracking effect by nonlinear adaptive decoupling control strategy from 06:00 a.m. to 20:00 p.m.

**Figure 18 sensors-23-08323-f018:**
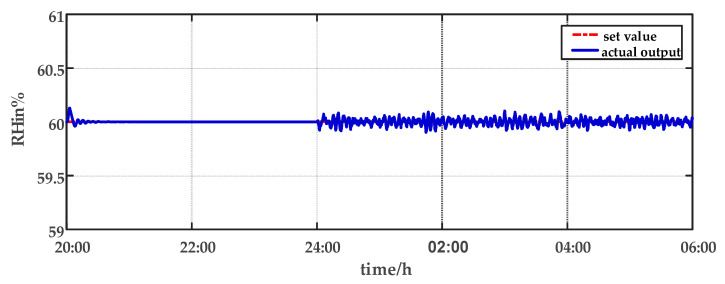
The effect of humidity by nonlinear adaptive decoupling control strategy in parameter uncertainty experiment and multi-disturbance experiment from 20:00 p.m. to 06:00 a.m.

**Figure 19 sensors-23-08323-f019:**
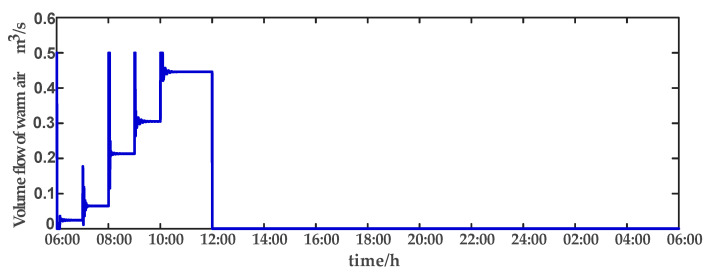
Volume flow rate of warm air ($u_{1}$) under the nonlinear adaptive decoupling control strategy.

**Figure 20 sensors-23-08323-f020:**
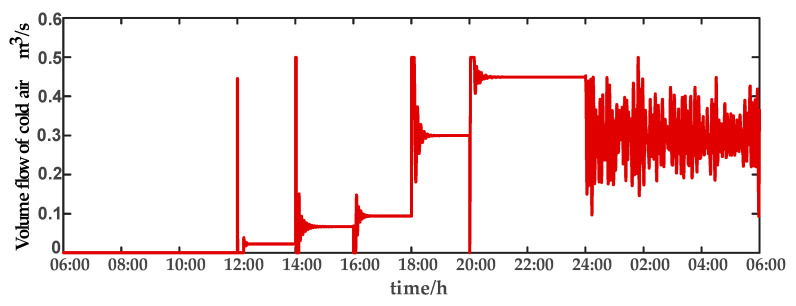
Volume flow rate of cold air ($u_{2}$) under the nonlinear adaptive decoupling control strategy.

**Figure 21 sensors-23-08323-f021:**
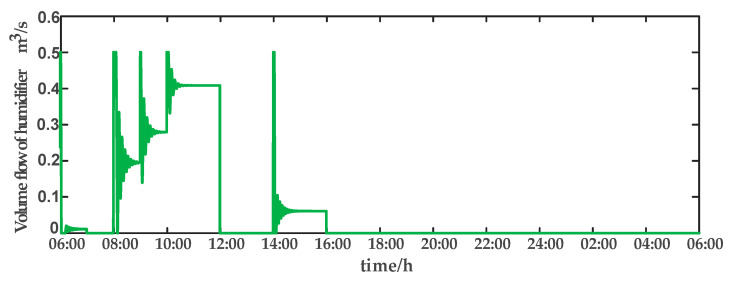
Volume flow rate of humidifier ($u_{3}$) under the nonlinear adaptive decoupling control strategy.

**Figure 22 sensors-23-08323-f022:**
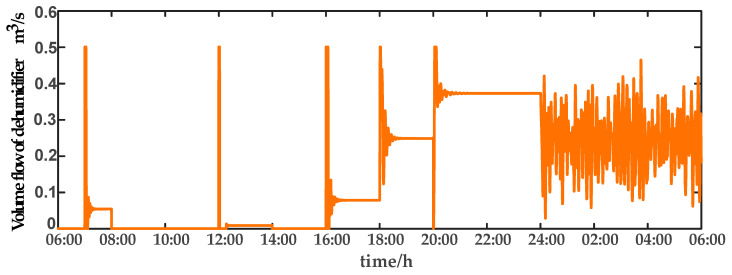
Volume flow rate of dehumidifier ($u_{4}$) under the nonlinear adaptive decoupling control strategy.

**Table 1 sensors-23-08323-t001:** Parameter meaning of temperature and humidity model of plant factory.

Parameter	Meaning	Value Range	Unit
$k_{1}$	The illumination utilization coefficient of artificial light sources	0.97	
$k_{2}$	The special permissive coefficient of artificial light sources	1	
$P_{i}$	Power of a single artificial light source	50–300	w
n	Total quantity of artificial light sources	0–8	
$t_{w}$	Working time of artificial light sources	0–4	h
$R_{n}$	Net radiation intensity of crop canopy	0–350	$w/m^{2}$
K	The decay coefficient of LED lamps	0–1	
γ	Thermometer constant	0.0646	kPa/°C
$e_{0}$	The saturated water vapor pressure of the air in the factory at 0 °C	0.6107	kPa
$T_{P}$	Leaf temperature	15–28	°C
LAI	Crop leaf area index	0.125	
$A_{P}$	Leaf area	12	$m^{2}$
V	Internal volume of a plant factory	288	$m^{3}$
A	Internal ground area of a plant factory	96	$m^{2}$
$\rho_{a}$	Density of air in a plant factory	1.199	$kg/m^{3}$
$C_{ap}$	Specific heat at constant pressure in a plant factory	1.009	kJ/(kg·°C)
P	Standard atmospheric pressure	101,325	Pa
$P_{0}$	The water vapor partial pressure at different temperatures		Pa
$c_{r}$	Heat transfer coefficient of wall	0.002–0.003	
$A_{b}$	Internal wall area of a plant factory	132	$m^{2}$
$t_{b}$	Temperature of inner wall of a plant factory	12–28	°C
$q_{f}$	Fresh air flow rate for personnel entry and exit, ventilation	0–1	$m^{3}/s$
$H_{hw}$	The humidity ratio during the supply of hot air	14–19	$g/m^{3}$
$H_{ca}$	The humidity ratio during the supply of cold air	16–21	$g/m^{3}$
$H_{dh}$	The humidity ratio of the supply air of the dehumidifier	13.05	$g/m^{3}$
$H_{hum}$	The humidity ratio of the supply air of the humidifier	25	$g/m^{3}$
$T_{hw}$	Hot air supply temperature	25–35	°C
$T_{ca}$	Cold air supply temperature	10–20	°C
$T_{hum}$	Temperature during the operation of the humidifier	20–25	°C
$T_{dh}$	Temperature during the operation of the dehumidifier	15–30	°C

**Table 2 sensors-23-08323-t002:** 24 h Setting values of temperature and humidity for plant factory.

Time	Temperature (°C)	Relative Humidity (%)
00:00–06:00	15	60
06:00–07:00	20	65
07:00–08:00	20	60
08:00–09:00	26	70
09:00–10:00	28	70
10:00–12:00	30	70
12:00–14:00	28	65
14:00–16:00	23	70
16:00–18:00	20	60
18:00–24:00	15	60

**Table 3 sensors-23-08323-t003:** Comparison of control performance between traditional PID method and nonlinear adaptive decoupling control method.

Methods	Temperature Error (°C)		Humidity Error (kg/kg)	
	Mean	Standard	Mean	Standard
Conventional PID	0.3615	0.8425	0.1475	0.4410
Nonlinear adaptive decoupling control	0.1655	0.6665	0.0221	0.1541

## Data Availability

Data are unavailable due to privacy.

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
