# Peer review of "High-Order Neural-Network-Based Multi-Model Nonlinear Adaptive Decoupling Control for Microclimate Environment of Plant Factory"

_sensors, 2023, doi:10.3390/s23198323_

Round 1
Reviewer 1 Report
Could also provide information on the advantages and idsadvantages of the linear and non-linear models? You can refer to and cite the following literature:
Akin, M., Eyduran, S. P., Eyduran, E., & Reed, B. M. (2020). Analysis of macro nutrient related growth responses using multivariate adaptive regression splines. Plant Cell, Tissue and Organ Culture (PCTOC), 140, 661-670.
Minor editing of English language required.
Author Response
We appreciate your invaluable recommendations. Please find the detailed response provided in the document attached.

Reviewer 2 Report
The manuscript "Higher-order Neural Network-based Multi-model Nonlinear Adaptive Decoupling Control for Microclimate Environment of Plant Factory" Is well-designed and has very good, interesting and novel results that might be interested in the experts. However, there are some minor items that should be improved before acceptance, as follows:
1- The text need to be edited by native English expert. There are some typos and grammatical mistakes.
2- All the figures are well presented, but it is recommended to revise them with high resolution.
After consideration of the above-mentioned items, the manuscript can be accepted
It is needed, the text will be edited by native English experts. There are some typos and grammatical mistakes that should be corrected.
Author Response

(The authors gave the same response as above.)

Reviewer 3 Report
1.This well-written manuscript presents intriguing research findings, although there is room for some minor improvements.
2.The abstract is satisfactory, but it may not sufficiently engage readers to continue reading the entire manuscript. Please provide a clear description of the manuscript's content.
3.The equations in the manuscript are well done and scientifically sound, contributing to the repeatability of the manuscript’s proposals. Each equation in the manuscript should be explicitly introduced in the main text (e.g., Equations 9,15, etc... are not yet introduced). In sections 2 and 3, I recommend that the authors review their equations and notations to ensure their accuracy, as errors have been identified (for example, the notations in Table 1 do not match the notations in Equation(22)and Equation(23), which can easily cause confusion for readers).
4. The author should carefully check the structure of the manuscript, as the sudden introduction of section 3.2 may cause confusion among readers (e.g., in Line 315).
The referenced formula and content should correspond (for example, the formula referenced in line 282 clearly does not correspond to the content). The variables in the equation have not been detailed (for example, in line 319). The method mentioned in line 342 is not reflected in the text.
5. In section 4, please emphasize your main achievements and their contributions to the development, environment, as well as other aspects of plant factory and smart agriculture.
Minor editing of English language required
Author Response

(The authors gave the same response as above.)

Reviewer 4 Report
The authors present an interesting paper about applying Higher-order Neural Networks and Nonlinear Adaptive Decoupling Control to control an indoor environment to support crop production. However, they should improve or better explain some issues, namely:
1.- The writing style used with many of the equations incorporated in the text, instead of highlighting them separately, makes it very difficult for the reader to read and understand.
2.- Often some of the parameters used in the equations are not identified or their meaning is not stated. see for example eq(5) and (6).
3.-The use of intermediate functional diagrams would allow the user to better follow the authors' reasoning in their mathematical demonstrations.
4.- Since the proposed method is "almost perfect" following the change in setpoints, why in Fig 17, do the transitions present characteristics similar to the "Gibs phenomenon".
5.- In terms of temperature, we are talking about variations of a tenth of a degree (Fig6), and in terms of RH, we are talking about 0.25% (fig9). These variations are relevant to the process in terms of real and practical testing. Effectively, the proposed model presents a higher quality response, almost ideally following the set-point variation, but what is the difference in computational power required for the process? This difference is justified by the improvements presented in the examples used in the article.
6.- As presented in the paper, the method proposed by the authors presents the greatest differences between the "Volume flow rate of humidifier/dehumidifier" and "Volume flow rate cold/hot air", which translates into a longer operating period of the actuators, promoting greater energy expenditure, right?
Finally, congratulations on your work
Author Response

(The authors gave the same response as above.)
